# Behavioral Propagation Based on Passionate Psychology on Single Networks with Limited Contact

**DOI:** 10.3390/e25020303

**Published:** 2023-02-06

**Authors:** Siyuan Liu, Yang Tian, Xuzhen Zhu

**Affiliations:** State Key Laboratory of Networking and Switching Technology, Beijing University of Posts and Telecommunications, Beijing 100876, China

**Keywords:** complex networks, information propagation, limited contact network, passionate psychology

## Abstract

Passionate psychology behavior is a common behavior in everyday society but has been rarely studied on complex networks; so, it needs to be explored in more scenarios. In fact, the limited contact feature network will be closer to the real scene. In this paper, we study the influence of sensitive behavior and the heterogeneity of individual contact ability in a single-layer limited-contact network, and propose a single-layer model with limited contact that includes passionate psychology behaviors. Then, a generalized edge partition theory is used to study the information propagation mechanism of the model. Experimental results show that a cross-phase transition occurs. In this model, when individuals display positive passionate psychology behaviors, the final spreading scope will show a second-order continuous increase. When the individual exhibits negative sensitive behavior, the final spreading scope will show a first-order discontinuous increase In addition, heterogeneity in individuals’ limited contact capabilities alters the speed of information propagation and the pattern of global adoption. Eventually, the outcomes of the theoretic analysis match those of the simulations.

## 1. Introduction

In the field of complex network research, disease transmission [1,2], social transmission [3], resource division [4], computer virus transmission [5], etc. are research areas of many researchers, among which social transmission mainly focuses on behavioral transmission [6], information transmission [7], and emotional transmission [8] among people. These transmissions are the embodiment of some behaviors and habits of people in daily life and have high individual autonomy; so, they have attracted the attention of many researchers. They theoretically derived social transmission through formula derivation and also built complex networks for experimental verification. Besides, social transmission has a certain special reinforcement effect [9,10,11].

The most commonly used approach in the early stages of research is based on the threshold model for the memoryless Markov process [12,13]. In this threshold model, when the number of neighbor nodes adopting the behavior exceeds the predetermined adoption threshold, the behavior will be adopted [14,15]. Since the proportion of the initial seed number is little, the percolation theory can be used to estimate this proportion [16,17]. When the adoption threshold is fixed, the change in the average degree scale will cause the saddle point bifurcation, causing the final adoption scale to initially increase continuously and then discontinuously decrease as the average degree of the network increases [18,19]. It is found that in the threshold model, multiple factors such as initial seed number [18], clustering coefficient [20], and community structure [21] will affect the propagation process.

Limited contact ability and different behaviors of individuals are important factors affecting information propagation. Because people’s social contact range will be affected by personal personality, work, social ability, and other factors in real life, each person often has his own unique social circle. In this social range, Golder et al. [22] proposed that people do not communicate frequently with all their friends and often only communicate with a few friends at a time. For a brief period of time, scientists typically co-author publications with a small number of partners [23,24]. In addition, a non-Markovian model is proposed by Wang et al. [25] to describe how limited contact capacity affects the spread of information and discovered that as contact capacity enhances, the network becomes more susceptible to the spread of behavior. Therefore, the limited contact ability for information propagation must be investigated.

Through extensive investigations, we uncover that a behavioral model of passionate psychology [26,27] can influence information propagation mechanisms. When Dodds et al. [28] proposed the tent threshold model and the changing square threshold model when studying the propagation of individual behavior, and then used the binary state model, they found that the above models would cause confusion in Markovian social transmission. The characteristic of this model is that the threshold will first rise and then fall with the adoption of neighbor nodes. This threshold model is proposed based on the snob effect and has certain reference significance. In fact, people’s enthusiasm for some behaviors and things tends to rise first and then remain unchanged after rising to a certain range. We call this behavior the like effect. In popular culture right now, there are a lot of these effects happening from time to time. For example, individuals often listen to their favorite songs repeatedly and spread their favorite songs to people around them. The research on this kind of behavior in the field of social communication is still very rare, so it has a strong research value.

Since this passionate psychology behavior of non-Markovian social communication has not been studied systematically, this paper proposes a social transmission model to study the propagation dynamics of this passionate psychology behavior in a single-layer limited-contact network—that is, each individual exhibits passionate psychology behavior and can only send information to a fixed number of neighbors. The adoption threshold in the form of a rectangular trapezium is used to describe the passionate psychology, first rising and then maintaining the horizontal line. The adoption threshold model is used to simulate the strengthening effect of non-redundant information on information propagation, and a quantitative verification and analysis based on generalized edge partition theory is proposed. Through rigorous theoretical analysis and extensive experimental simulations, the results show that the theoretical analysis values can fit well with the experimental simulation values.

## 2. Materials and Methods

### 2.1. Model Descriptions

To investigate the effects of passionate psychology behaviors on information propagation on limited contact networks, we suggest a single-layer limited-contact network of *N* nodes. The network is an information propagation network. In this network, $k_{i}$ represents the degree of individual i, and for the degree $k_{i}$, the degree distribution is represented by P(k). In Figure 1, to represent behavior spreading in a single-layered network with limited contact capacity, the degree distribution P(k), and *N* individuals, we employ the susceptible–adopted–recovered (SAR) model [29,30]. It indicates that each individual will be in one of three states at any time: susceptible (*S*) state, adopted (*A*) state, or recovered (*R*) state. In subgraph (c), we denote the S-state in blue, the A-state in red, and the R-state in yellow. An individuals in the S-state has not adopted this behavior. An individual in A-state has adopted this behavior and can also pass the information on to its neighbors in S-state. The individual in the R-state gives up the behavior and stops passing it on.

First, we develop a single-layer limited-contact network. We use $f\left(k_{i}\right)$ to represent the contact capacity of the individual *i* with the Adopted state on the limited contact networks, where $k_{i}$ is the degree of *i*. The maximum number of neighbors an A-state individual can broadcast behavior information to is equal to $f\left(k_{i}\right)$. All neighbors can be transmitted information by the individual *i* with the A-state if $f\left(k_{i}\right)\geq k_{i}$. If $f\left(k_{i}\right)<k_{i}$, the individual *i* with A-state can only choose $f\left(k_{i}\right)$ of its neighbors at random for the propagation of information, and the information adoption probability equals the propagation probability λ for these chosen neighbors. If individual *i* in A-state encounters a neighbor individual *j* in S-state, the probability of individual *i* being infected is $\frac{\lambda \left(f\left(k_{i}\right)\right)}{k_{i}}$.

Second, we introduce passionate psychology behaviors in social communication. The mathematical function corresponding to the passionate psychology behavior is
(1)$$h\left(x,a\right)=\left\{\begin{array}{c} \frac{x}{a},\;0<x\leq a\\ 1,\;a<x\leq 1 \end{array}\right.$$
where *x* is the ratio between an individual’s received information and their degree. The parameter of each individual passionate psychology behavior is represented by the symbol *a*. When 0<x≤a, the adoption probability rises with *x*. When a<x<1, the adoption probability no longer changes with *x* and remains constant. This function illustrates that small *a* enhances the individual’s passionate psychology behavior. The individual’s passionate psychology behavior, however, weakens as the *a* value increases.

We use the well-known ScaleFree (SF) network model [31] and the Erds–Rnyi (ER) [32] network model as the physical foundation for our experiments. The variable m is used to calculate how much information a person has acquired. Individual I acquires information from a neighbor once more, increasing m by one.

The individual in the S-state may adopt the information with the probability $h(\frac{m}{k},a)$, in which *k* is the symbol of the degree of the individual in the S-state. The S-state individual *i* has the possibility to accept the information and change to the A-state. The information propagation process is not Markovian because we take the impact of non-redundant information propagation into account.

The model’s process for how information spreads is as follows:

A portion of the $\rho_{0}$ individuals are initially chosen to be the A-state individuals; at the same time, the other individuals are in the susceptible state. Individuals who have not obtained any information have received a total of m=0 pieces of information. Each adopted individual *i* chooses $f\left(k_{i}\right)$ neighbors randomly at each time step before transmitting the information to its chosen S-state neighbors with the probability λ. So, individual *j* may be infected with the probability $\frac{\lambda \left(f\left(k_{i}\right)\right)}{k_{i}}$. The amount of information pieces *m* increases by one when the A-state individual sends the behavior information to its neighbor *i*. Individual *i* cannot send this information to individual *j* after the transmission is successful. At the same time, the S-state individual may adopt the information with the probability $h(\frac{m}{k},a)$. The susceptible individual *i* has the opportunity of entering A-state or remaining in S-state. Following the information propagation, the adopted state individual *j*, with probability γ, may change into the recovered state individual but is unable to participate in the ensuing propagation. When there are no A-state individuals left, this propagation will come to an end.

### 2.2. Theory Analysis

Based on references, we analyze the model theory using an edge-based compartmental (EBC) approach. By comparing the proportional changes of various state individuals in single-layer networks, we theoretically evaluate the information propagation mechanism of single-layer networks with limited contacts.

According to the cavity theory, the individual *i* can gather information from its neighbors but cannot share information with others. Up to time *t*, the probability that individual *j* whose degree is $k_{j}$ does not send data to individual *i* is provided using the function $\theta_{k_{j}}\left(t\right)$. As a result, the probability that the individual *i* will not obtain the behavior information up to time *t* is
(2)$$\theta \left(t\right)=\sum_{k_{j}=0}\frac{k_{j}P\left(k_{j}\right)}{\langle k\rangle }\theta_{k_{j}}\left(t\right)$$
where $\frac{k_{j}P\left(k_{j}\right)}{\langle k\rangle }$ represents the probability that the individual *j* will be connected to *i*. The probability that the S-state individual *i* whose degree is $k_{i}$ obtains *m* pieces of information collectively by time *t* is
(3)$$\phi_{m}\left(k_{i},t\right)=\left(\begin{array}{c} k_{i}\\ m \end{array}\right)\theta {\left(t\right)}^{k_{i}-m}{\left[1-\theta (t)\right]}^{m}$$

With a probability of $\prod_{l=0}^{m}\left[1-h(\frac{l}{k},a)\right]$, the individual *i* with *m* pieces of information has not taken on the behavior and is still in the S-state. After collectively obtaining *m* pieces of information, by time *t*, the individual *i* still keeps the S-state with the probability
(4)$$\begin{array}{c} \tau_{X}\left(k_{i},m,t\right)=\sum_{m}^{k_{i}}\phi_{m}\left(k_{i},t\right)\prod_{l=0}^{m}\left[1-h(\frac{l}{k_{i}},a)\right]\\ =\sum_{m=0}^{\left\lfloor bk_{i}\right\rfloor }\phi_{m}\left(k_{i},t\right)\prod_{l=0}^{m}\left[1-\frac{l}{ak_{i}}\right] \end{array}$$

So, after having accumulated *m* pieces of information, up to time *t*, the probability that the susceptible individual *i* with the degree of $\vec{k_{i}}=\left(k_{i}\right)$ keeps the S-state is
(5)$$\begin{array}{c} s\left(\vec{k},t\right)=\left(1-\rho_{0}\right)\sum_{m=0}^{k_{i}}\phi_{m}\left(k_{i},t\right)\prod_{l=0}^{m}\left[1-h(\frac{l}{k_{i}},a)\right]=\left(1-\rho_{0}\right)\tau \left(k_{i},m,t\right) \end{array}$$

Up to time *t*, the aggregate information pieces after the S-state individual *i* do not change the state with the probability
(6)$$\eta =\sum_{k_{i}}P\left(k_{i}\right)\tau \left(k_{i},m,t\right)$$

As a result, at time *t*, the S-state individuals in the single-layer network with the proportion
(7)$$s\left(t\right)=\sum_{\vec{k}}P\left(\vec{k}\right)s\left(\vec{k},t\right)=\left(1-\rho_{0}\right)\eta$$

Given that each individual in this model is in one of three different states, the value of $\theta_{k_{j}}\left(t\right)$ can be represented as
(8)$$\theta_{k_{j}}\left(t\right)=\xi_{S,k_{j}}\left(t\right)+\xi_{A,k_{j}}\left(t\right)+\xi_{R,k_{j}}\left(t\right)$$
where $\xi_{S,k_{j}}\left(t\right)$, $\xi_{A,k_{j}}\left(t\right)$, and $\xi_{R,k_{j}}\left(t\right)$ represent the probability that the individual with degree $k_{j}$ is susceptible, adopted, or recovered and does not translate the behavior information to its neighbors, respectively.

The individual *i* cannot send information to other neighbor individuals because it is in the cavity state. Therefore, the susceptible individual *j* can only obtain information from its other $k_{j}-1$ neighbors. The individual *j* obtains *n* pieces of information from its neighboring individuals by time *t* with the probability $\zeta (k_{j}-1,n,t)$. The probability $\zeta (k_{j}-1,n,t)$ can be calculated by
(9)$$\begin{array}{c} \zeta \left(k_{j}-1,n,t\right)=\sum_{n}^{k_{j}-1}\phi_{n}\left(k_{j}-1,t\right)\prod_{l=0}^{n}\left[1-h(\frac{l}{k_{j}},a)\right]\\ =\sum_{n=0}^{\left\lfloor ak_{j}\right\rfloor }\phi_{n}\left(k_{j},t\right)\prod_{l=0}^{n}\left[1-\frac{l}{ak_{j}}\right] \end{array}$$

The individual *j* is still susceptible by time *t* with the probability
(10)$$\xi_{S,k_{j}}\left(t\right)=\left(1-\rho_{0}\right)\zeta \left(k_{j}-1,n,t\right)$$

Because of the limited contact capacity, an A-state individual *j* with degree $k_{j}$ chooses a neighbor to spread information with probability $\frac{f\left(k_{j}\right)}{k_{j}}$. Additionally, there is a λ chance that the information will pass across this edge. Consequently, the probability that an individual *j* will spread through an edge is $\frac{\lambda f(k_{j})}{k_{j}}$. $\frac{d\theta_{k_{j}}\left(t\right)}{dt}$ can be written as
(11)$$\frac{d\theta_{k_{j}}\left(t\right)}{dt}=-\frac{\lambda f(k_{j})}{k_{j}}\xi_{A,k_{j}}$$

The A-state individuals may become the R-state with the probability γ, and $\xi_{R,k_{j}}\left(t\right)$ can be represented as
(12)$$\frac{d\xi_{R,k_{j}}\left(t\right)}{dt}=-\gamma \xi_{A,k_{j}}\left(t\right)\left(1-\frac{\lambda f(k_{j})}{k_{j}}\right)$$

Given the initial circumstances $\theta_{k_{j}}\left(0\right)=1,\xi_{R,k_{j}}\left(0\right)=0$, we can obtain
(13)$$\frac{d\theta_{k_{j}}\left(t\right)}{dt}=-\frac{\lambda f(k_{j})}{k_{j}}\left[\theta_{k_{j}}\left(t\right)-\xi_{S,k_{j}}\left(t\right)\right]+\gamma \left[1-\theta_{k_{j}}\left(t\right)\right]\left[1-\frac{\lambda f(k_{j})}{k_{j}}\right]$$

When t→∞, we can obtain $\theta_{k_{j}}\left(t\right)$ from Equation (Equation 16)—that is,
(14)$$\theta_{k_{j}}\left(\infty \right)=\frac{\xi_{S,k_{j}}\left(t\right)+\gamma \left[\frac{k_{j}}{\lambda f(k_{j})}-1\right]}{1+\gamma \left[\frac{k_{j}}{\lambda f(k_{j})}-1\right]}$$

Substituting $\theta_{k_{j}}\left(\infty \right)$ into Equation (Equation 1), we can obtain
(15)$$\theta \left(\infty \right)=\sum_{k_{j}=0}\frac{k_{j}P\left(k_{j}\right)}{\langle k\rangle }\theta_{k_{j}}\left(\infty \right)=f\left(\theta \left(\infty \right)\right)$$

For simplicity, we can rewrite θ(∞) as
(16)θ(∞)=f(θ(∞))

There is a discontinuous growth pattern when Equation (Equation 14) is tangent to Equation (Equation 15) with θ(∞)<1. From the following equation, we can determine what the critical conditions are
(17)$$\frac{\partial f(\theta (\infty ))}{\partial \theta (\infty )}=1$$

### 2.3. Parameter Settings

The quantity of individuals on a single-layer SF network and ER network are set as $10^{4}$ and the average degree of individuals is set as $\langle k\rangle =10$. In the single-layer ER network, the degree distribution follows the Poisson distribution $p\left(k\right)=e^{-\langle k\rangle }\frac{{\langle k\rangle }^{k}}{k!}$. In the single-layer SF network, the degree distribution follows power-law distribution $p\left(k\right)=\xi k^{-v}$, where *v* presents the degree exponent and $\xi =1/\sum_{k}k^{-v}$. Additionally, a very tiny percentage of seeds ($\rho_{0}$ = 0.0001) and probability γ = 1.0 both support R-state transitions in A-state individuals.

There are no A-state individuals in the network when it reaches a stable state as a whole. The proportion of individuals in the R-state is set at R(∞) to calculate the final spreading scope. Additionally, the experimental result can be phrased as follows to determine the critical condition from the simulations:(18)$$\chi =\frac{\langle {\left(R\left(\infty \right)-\langle R\left(\infty \right)\rangle \right)}^{2}\rangle }{{\langle R\left(\infty \right)\rangle }^{2}}$$
where R(∞) denotes the final information spreading scope and $\langle \ldots \rangle$ represents ensemble mean. The critical points of the final adoption scope are implied by the χ peak values.

## 3. Simulation and Discussion

Based on the fundamental parameters in the networks, we simulate and analyze our propagation model using experiments with information propagation on the single-layer limited ER network and SF network, respectively.

### 3.1. Information Propagation on Single-Layer Limited ER Network

In Figure 2, we can obtain the contact capacity that can influence the evolution of the A-state and the outbreak time of information propagation. By comparing (a) and (b), we can deduce that the evolution time cost continuously reduces from 11 to 10, while R(∞), which denotes that at the same step *t*, the final spreading scope increases. The phenomenon demonstrates that increasing an individual’s capacity for contact (i.e., the number of contact neighbors) can hasten the dissemination of information across a single-layer network.

Figure 3a (c=6) and Figure 3c (c=10) show that as λ increases, the final spreading scope R(∞) increases to global spreading. Furthermore, Figure 2a,c also show the influence of the passionate psychology parameter *a* on the phase transition. In subgraph (a), when an individual shows a positive passionate psychology behavior—e.g., a=0.2—the pattern of R(∞) shows a second-order continuous growth, suggesting that even with a small λ, a positive passionate psychology can promote information propagation. When a=0.5, the individual shows a passive passionate psychology behavior, and R(∞) also shows a first-order discontinuous growth. When a=0.8, however, the spread of information does not break out. In subgraph (c), R(∞) shows a second-order continuous growth at a=0.2. R(∞) similarly exhibits a first-order discontinuous growth at a=0.5 and a=0.8. The phenomenon illustrates that the limited contact capacity *c* changes the pattern of information spreading, such as a=0.8 in subgraphs (a) and (c).

The relative error and crucial information spreading possibilities of Figure 3a,c separately, are shown in Figure 3b,d. The information about global adoption will emerge from the deviation of information propagation, which is represented by the highest values of relative error chi. As individual capacity to make contact with one another increases, the process of information propagation will begin earlier. Additionally, the results of the simulation (symbols) agree with our theoretical analysis (lines).

Figure 4 illustrates that the combined impact of the parameter plane (λ,a) on R(∞) for the single-layer ER network. The impacts of (λ,a) on the information spreading scope are shown in Figure 4a,b, respectively. Three spreading phenomena are both shown in subgraphs (a) and (b) as the value of λ increases. After that, the diagram can then be divided into three regions. From region I, the continuous phase transition of R(∞) with second order is visible. In region II, the growing R(∞) pattern displays a first-order discontinuous phase transition as λ increases. The growth of R(∞) does not appear in region III, meaning that there is no information outbreak in this region. In subgraph (b), two spreading phenomena appear as the value of λ increases. Region I also shows the second-order continuous phase transition of R(∞). The discontinuous phase transition of R(∞) with first order is visible from region II. The limited contact capacity of individuals can affect both the information spreading and the transition from a second-order continuous phase transition to a first-order discontinuous phase transition.

### 3.2. Information Propagation on Single-Layer Limited SF Network

Figure 5 shows the impact of unit spreading possibility λ and passionate psychology behaviors *a* on the information final spreading scope for the single-layer SF network with limited contact. The same individual contact capacity (c=10) is applied to each subgraph. Figure 5a,c demonstrate that as λ rises, the final adoption scope R(∞) increases until it reaches global propagation. In subgraph (a) (v=2), the growing pattern of final spreading scope exhibits a continuous phase transition with second order when a=0.2. When a=0.5, the growth of R(∞) shows a first-order discontinuous phase transition. When a=0.8, however, the spread of information does not break out. Subgraph (c) shows a different propagation phenomenon at different parameters *v*. In subgraph (c) (v=4), the growing pattern of final spreading scope exhibits a continuous phase transition with second order when a=0.2. However, when a=0.5 and a=0.8, the growth of R(∞) shows a first-order discontinuous phase transition. However, when comparing subgraphs (a) and (c), it can be observed that a relatively small degree distribution exponent (v=4) can drive information propagation and global adoption. The relative error and crucial information spreading possibilities of Figure 5a,c, separately, are shown in Figure 5b,d. Additionally, our theoretical analyses (lines) match the outcomes of the simulation (symbols).

The combined impact of the parameter plane (λ,a) on R(∞) for the single-layer SF network is verified in Figure 6. The impacts of (λ,a) on the information spreading scope are shown in Figure 6a,b, respectively. In subgraph (a), three spreading phenomena appear as the value of λ increases. The diagram can then be split into three regions. The continuous phase transition of R(∞) with second order is visible from region I. As increases in region II, the growing R(∞) pattern displays a first-order discontinuous phase transition. The region above the dotted line shows that the growth of R(∞) does not appear, meaning that there is no information outbreak in the region. In subgraph (b), two spreading phenomena appear as the value of λ increases. The continuous phase transition of R(∞) with second order is also visible from region I. The discontinuous phase transition of R(∞) with first order is visible from region II. In region III, R(∞) does not grow, meaning that there is no information outbreak in the region. The degree of individual heterogeneity distribution in the single-layer SF networks affects both the spread of information and the transformation from a second-order continuous phase transition to a first-order discontinuous phase transition.

## 4. Conclusions

In this paper, the effect of passionate psychology behavior on a single-layer limited contact network is examined. We propose a novel propagation effect of propagation behavior on limited contact networks—that is, the single-layer limited-contact network—with passionate psychology models. The effect of information propagation from simulations and theoretical studies is then examined using the developed edge-based compartmental (EBC) theory. Through numerous experiments, our simulation results are in good agreement with theoretical findings.

On the single-layer ER networks, we discovered that changing the contact capacity *c* and passionate psychology parameter *a* can have an impact on the phenomena of information spreading. On single-layer ER networks, there is a crossover phenomenon where R(∞) can grow either continuously or discontinuously as λ increases. Specifically, when individuals possess weak contact capacity, the information propagation process shows a continuous increase through a positive passionate psychology behavior. However, a passive passionate psychology behavior makes the information propagation show a discontinuous increase until the information propagation disappears. When individuals possess strong contact capacity, the information propagation process also shows a continuous increase through a positive passionate psychology behavior. The information propagation shows a discontinuous increase by a passive passionate psychology behavior. Furthermore, the strong contact capacity of individuals has driven the global adoption of information all the time. On the single-layer SF networks, we found that the passionate psychology parameter *a* and degree distribution exponent *v* can affect information spreading. When SF network shows strong degree distribution heterogeneity, the information propagation process shows a continuous increase through a positive passionate psychology behavior. However, a passive passionate psychology behavior makes the information propagation show a discontinuous increase until the information propagation disappears. When the SF network shows weak degree distribution heterogeneity, the information propagation process also shows a continuous increase through a positive passionate psychology behavior. The information propagation shows a discontinuous increase by a passive passionate psychology behavior.

For passionate psychology behaviors, we explore the information propagation mechanism of the new feature behavior. However, in the real world, due to various types of networks, different networks can lead to different propagation phenomena; meanwhile, the impact of passionate psychology behavior on other types of networks needs further research.

## Figures and Tables

**Figure 1 entropy-25-00303-f001:**
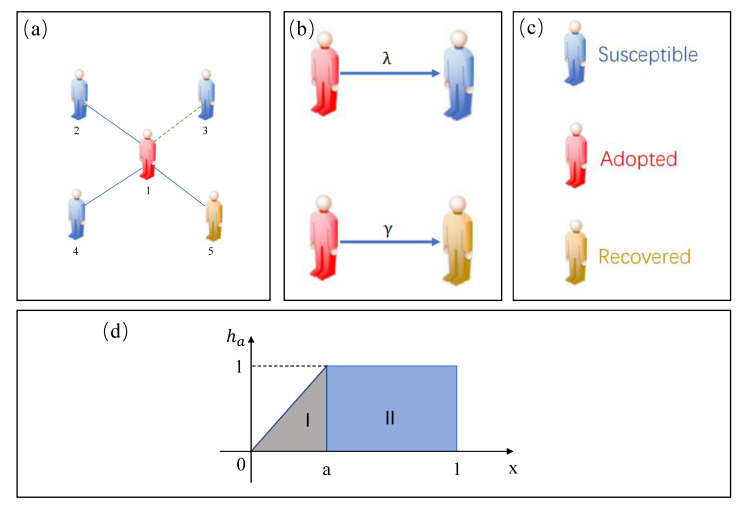
(**a**) A schematic diagram of information spreading in a single-layer complex network. Individual 1 is in the adopted state; individuals 2, 3, and 4 are in the susceptible state; and individual 5 is in the recovered state. The blue solid line indicates that the information is still in the state not propagated from this path; the green dashed line indicates that at the current time, the adopted state has passed information to its neighbors through this path. (**b**) Illustrates the probability λ of information passing from the A-state to the S-state and the probability γ of the A-state becoming the R-state. (**c**) Illustrates that each color corresponds to a different state. (**d**) Illustrates a probabilistic graphical representation of passionate psychology behavior. In area I, when 0<x≤a, the adoption probability rises with *x*.In area II, when a<x<1, the adoption probability no longer changes with *x* and remains constant.

**Figure 2 entropy-25-00303-f002:**
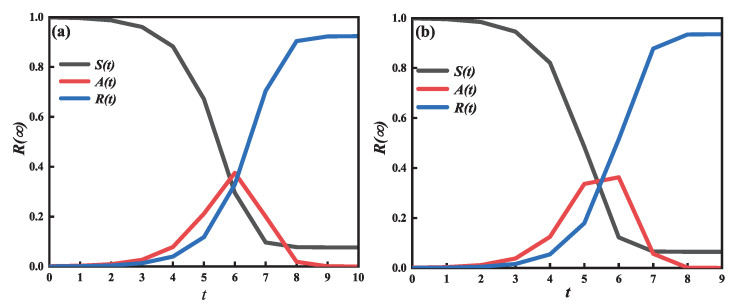
Individual density has fluctuated over time in various states. With the number of contact neighbors increasing from c=6 in subgraph (**a**) and c=10 in subgraph (**b**), for the same unit spreading probability, in subgraph (**a**), the rumor propagation costs 11 steps, but in subgraph (**b**) it costs 10 steps. By contrasting two subgraphs, it can be shown that a reinforcing contact capacity of individuals promotes the information spreading. The fundamental variables are a=0.2, λ=0.5, and $\rho_{0}=0.0001$.

**Figure 3 entropy-25-00303-f003:**
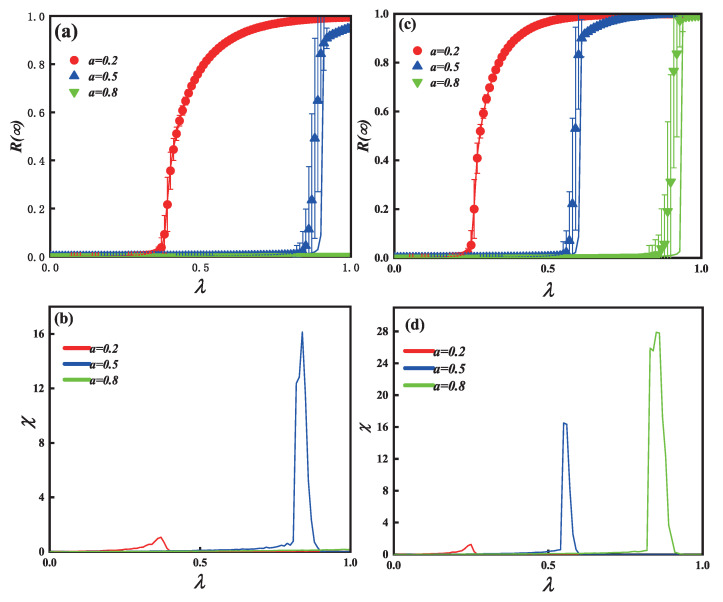
On the single-layer ER networks, the influence of limited contact capacity *c* and passionate psychology parameter *a* on the final information spreading scope R(∞). The impact of parameter *a* on the final information spreading scope R(∞) versus the propagation probability λ are depicted in subgraphs (**a**) (*c* = 6) and (**c**) (*c* = 10). Subgraphs (**b**,**d**) show the error between the simulation and theoretical value. Other parameters are $\rho_{0}=0.0001$ and γ = 1.0.

**Figure 4 entropy-25-00303-f004:**
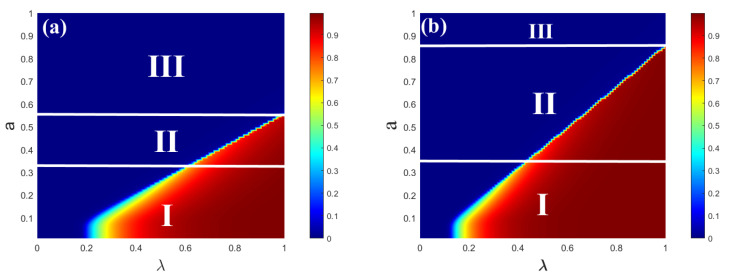
The combined roles of all passionate psychology parameter *a* and unit propagation probability on each individual’s eventual spreading size for a single-layer ER network. The impacts of subgraphs (**a**) (c=6) and (**b**) (c=10) on the final spreading scope are shown with the varying contact capacity of individuals. In subgraph (**a**), in region I, the growing R(∞) exhibits a second-order continuous phase transition. In region II, however, the growing R(∞) exhibits a first-order discontinuous phase transition. In region III, R(∞) does not grow, indicating that there is no information outbreak in the region. In subgraph (**b**), in region I, the growing R(∞) also exhibits a second-order continuous phase transition. In the same way, in region II, the growing R(∞) exhibits a first-order discontinuous phase transition. In region III, R(∞) does not grow, indicating that there is no information outbreak in the region. Other parameters are $\rho_{0}=0.0001$ and γ = 1.0.

**Figure 5 entropy-25-00303-f005:**
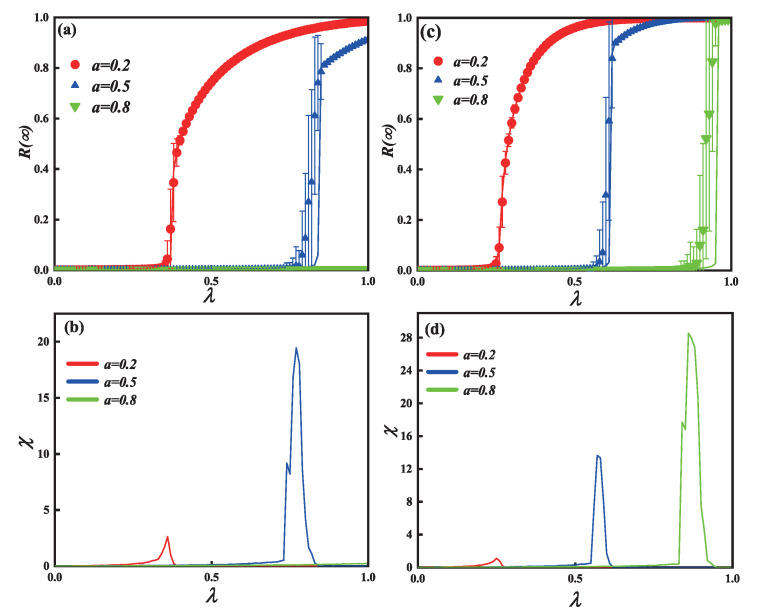
The impact of passionate psychology parameter *a* and unit propagation probability on each individual’s eventual spreading scope for the single-layer SF network. For each subgraph, the same contact capacity of individuals (c=10) is used. Additionally, the vertical subgraphs use the same degree distribution exponent, with subgraphs (**a**) to (**c**) corresponding to v=2 and v=4, accordingly. The impact on the final spreading size with unit propagation probability λ are presented in subgraphs (**a**,**c**). Subgraphs (**b**,**d**) show the error between the simulation and theoretical value. Other parameters are $\rho_{0}=0.0001$ and γ = 1.0.

**Figure 6 entropy-25-00303-f006:**
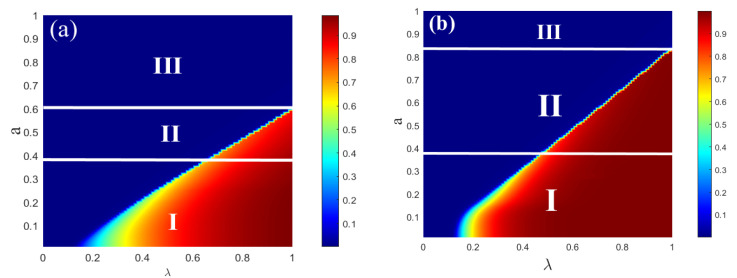
The combined effect of the passionate psychology parameter *a* and the unit propagation probability on the final spreading scope for the single-layer SF network. Subgraphs (**a**,**b**) show the impacts of (λ,a) on the final adoption scope with v=2 and v=4, respectively. In subgraph (**a**), in region I, the pattern of the final scope R(∞) grows continuously in the second-order phase transition. The change in R(∞) rises discontinuously in the phase transition with the first order of region II. Furthermore, region III above the dotted line shows that R(∞) does not grow, indicating that the information does not burst. In subgraph (**b**), in region I, the pattern of the final scope R(∞) also grows continuously in the second-order phase transition. The change in R(∞) rises discontinuously in the phase transition with the first order of region II. In region III, R(∞) does not grow, indicating that there is no information outbreak in the region. Other parameters are $\rho_{0}=0.0001$, γ = 1.0, and *c* = 10.

## Data Availability

All data are presented in main text.

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
