# Peer review of "Behavioral Propagation Based on Passionate Psychology on Single Networks with Limited Contact"

_entropy, 2023, doi:10.3390/e25020303_

Round 1
Reviewer 1 Report
In this paper, the authors propose a single-layer network with limited contact and explore the influence of people's passion psychology behaviour on information dissemination on this network. They considered the model based on the single-layer network with limited contact and proposed an edge partition theory to explain the propagation mechanism. There is no doubt that it is an exciting direction and research point. However, there are still some problems to be solved in this article.
1. What problem does the author want to solve? Just proposed a model?
2. How are the connections between the various nodes generated, random?
3. What do variable b in figure 3 and figure 5 represent? Is it best to describe it in the text?
4. The authors have chosen ER and SF networks as simulation networks in the paper. What are the characteristics and differences between these two networks, and why were they chosen?
5. Please check all expressions and promote the quality of the language.
Reviewer 2 Report
This paper explores the effect of passion psychology behaviors on information propagation in a single layer network with limited contact. And the propagation phenomenon is analyzed in ER and SF networks using edge division theory, respectively. The study is very interesting, but has several problems as follows:
1. The author lacks an analysis of the characteristic of the network model (ER and SF).
2. It should be interesting to analyze the cases in which the effects of limited contact capacity for SF networks.
3. In Figure 4, 6, what factors cause the change in the junction point between the second-order continuous phase transition and the first-order discontinuous phase transition, and how that change is explained?
4. In the INTRODUCTION section, the authors should introduce the motivations and the challenges faced by the research in this paper in detail.
5. The English writing of manuscript need to be strengthened.
